# A Review of Natural Fiber-Based Filaments for 3D Printing: Filament Fabrication and Characterization

**DOI:** 10.3390/ma16114052

**Published:** 2023-05-29

**Authors:** Mohd Nazri Ahmad, Mohamad Ridzwan Ishak, Mastura Mohammad Taha, Faizal Mustapha, Zulkiflle Leman

**Affiliations:** 1Department of Aerospace Engineering, Faculty of Engineering, Universiti Putra Malaysia, Serdang 43400, Selangor, Malaysia; faizalms@upm.edu.my; 2Aerospace Malaysia Research Centre (AMRC), Universiti Putra Malaysia, Serdang 43400, Selangor, Malaysia; 3Laboratory of Biocomposite Technology, Institute of Tropical Forestry and Forest Products (INTROP), Universiti Putra Malaysia, Serdang 43400, Selangor, Malaysia; 4Faculty of Mechanical and Manufacturing Engineering Technology, Universiti Teknikal Malaysia Melaka, Hang Tuah Jaya, Durian Tunggal 76100, Melaka, Malaysia; mastura.taha@utem.edu.my; 5Department of Mechanical and Manufacturing Engineering, Faculty of Engineering, Universiti Putra Malaysia, Serdang 43400, Selangor, Malaysia; zleman@upm.edu.my; 6Advanced Engineering Materials and Composites Research Centre, Faculty of Engineering, Universiti Putra Malaysia, Serdang 43400, Selangor, Malaysia; 7Centre of Smart System and Innovative Design, Universiti Teknikal Malaysia Melaka, Hang Tuah Jaya, Durian Tunggal 76100, Melaka, Malaysia

**Keywords:** natural fiber-based filaments, 3D printing, mechanical properties, dimension stability, surface quality, extrusion, fused deposition modeling

## Abstract

Today, additive manufacturing (AM) is the most recent technology used to produce detailed and complexly built parts for a variety of applications. The most emphasis has been given to fused deposition modeling (FDM) in the development and manufacturing fields. Natural fibers have received attention in the area of 3D printing to be employed as bio-filters with thermoplastics, which have prompted an effort for more ecologically acceptable methods of manufacturing. The development of natural fiber composite filaments for FDM requires meticulous methods and in-depth knowledge of the properties of natural fibers and their matrices. Thus, this paper reviews natural fiber-based 3D printing filaments. It covers the fabrication method and characterization of thermoplastic materials blended with natural fiber-produced wire filament. The characterization of wire filament includes the mechanical properties, dimension stability, morphological study, and surface quality. There is also a discussion of the difficulties in developing a natural fiber composite filament. Last but not least, the prospects of natural fiber-based filaments for FDM 3D printing are also discussed. It is hoped that, after reading this article, readers will have enough knowledge regarding how natural fiber composite filament for FDM is created.

## 1. Introduction

Rapid prototyping, often known as additive manufacturing (AM), is the method of fabricating layers of materials to create accurate parts or products [1]. It can also be defined as the process of creating a three-dimensional part from a digital three-dimensional model or CAD model. It was created at the end of the 1980s and is now a popular manufacturing technique [2]. Due to the simplicity of goods manufacturing, various AM techniques are currently widely used in industrial and home applications such as for photo frames, tripods, tools, souvenirs, gift accessories, and toys. Plenty of applications are restricted to smaller sizes that reduce the amount of waste and utilize energy without the need for larger machines [3]. AM is now being used in a variety of industries, including the biomedical, ceramic, plastic, and composite sectors, and it is a current area of interest for bio-printing, which includes the 3D printing of cardiac valves [4,5,6,7].

There are several types of AM technology, including material extrusion, direct energy deposition, powder bed fusion, sheet lamination, material jetting, binder jetting, and VAT photopolymerization [8]. For example, a model is built layer by layer using a process known as VAT polymerization, which uses a VAT of liquid photopolymer resin. Meanwhile, the sheet lamination process consists of UAM and LOM. DMLS, EBM, SHS, SLM, and SLS are some of the popular printing methods utilized in the powder bed fusion process. A powder-based material and a binder are the two ingredients used in the binder jetting method. In addition, FDM, otherwise known as FFF, was a widely used method for extruding materials. Layers of heated material are deposited after being drawn via a nozzle, where they are heated. FDM is also the most widely used due to its straightforwardness, versatility, rapid processing, affordable prices, dependability, low waste, variety of materials, and capacity to work with novel materials. The American firm (Stratasys), one of the market’s biggest players in 3D printing, invented the FDM technology in 1989. About 40% of the global market is accounted for by FDM technology [8]. The technology is intended for low-volume production; its benefits over traditional methods such as injection molding or extrusion are the decreased energy usage and reduced waste index [9,10]. Polymers, metals, and composites were all created for use in FDM; however, several of these materials are dangerous for both people and the natural environment because they produce organic substances that are volatile [11,12,13,14]. Polylactic acid (PLA), polypropylene (PP), polycarbonate (PC), and acrylonitrile butadiene styrene (ABS) are the most frequently used raw materials for 3D printing (FDM) [15,16,17]. FDM 3D printing is currently popular across a wide range of sectors, such as the manufacturing of consumer goods and the automotive, biomedical, and smart textile industries, etc. They employ FDM because it enhances the development of their products, the prototype process, and the manufacturing process. The potential of 3D printers to produce prosthetic human teeth was demonstrated by Ahmad et al. [18] in their study. Figure 1 shows a draw gear, a print head, a nozzle, a printing bed, and a spool of filament as an illustration of the basic FDM parts.

Natural fibers are widely available and found in many different varieties. Jute, kenaf, bamboo, dates, sisal, oil palm and sugar palms, pineapple, banana, and sisal are a few examples. Fibers such as rice straw, pineapple, wood, sugarcane, coir, and hemp have been used as reinforcements in polymer matrices by several studies to improve the performance of FDM-printed parts [20,21,22,23,24,25]. Typically, wood, seeds, grass, leafy, bast, and stalks are used to extract cellulose. Synthetic fiber is divided into two categories: organic fiber and inorganic fiber. Aramid, PE, and aromatic polyester compose organic fiber. Silica carbide, glass, boron, and carbon are examples of inorganic fibers [26]. As indicated in Table 1, the mechanical performance of each natural fiber varies. Oliveira et al. [27] found that composites that use natural cotton fiber as fillers had less of an impact on the environment. Overall, the eco-efficiency of the cotton-based composite materials also exhibited improvement. Because of their hierarchy-based woven structures, bamboo fibers have larger proportions of fiber displacement responses and matrix failures and significantly lower proportions of fracture features [28]. As a result, natural fibers have commonly been used as additions in 3D printing filaments [29]. Additionally, it is becoming more prevalent in a variety of industrial sectors, including construction and the automotive industry, and in thermally insulated composites and sound-absorbing materials [30].

This article reviews the natural fiber-based filaments for 3D printing. It discusses the fabrication process and the characterization of wire filament that is fabricated from thermoplastic material mixed with natural fiber. The characterization of wire filament includes its mechanical properties, dimension stability, morphological study, and surface quality. In addition, the challenges in developing a fiber composite filament are also discussed. Finally, the future perspectives of natural fiber-based filaments for FDM 3D printing are also presented.

## 2. Material Preparation

There are several animals, minerals, and plant sources that generate natural fibers [33]. As shown in Figure 2, they are made of cellulose embedded in a hemicellulose and lignin matrix [34]. Natural fibers absorb moisture, which causes dimensional changes in the composite and dissolves interfacial adhesion. The hydrophilic nature of natural fibers and the hydrophobic nature of the polymer matrix combine to create this problem, which may result in interfacial incompatibility and wettability. Chemically treated fibers have a better surface roughness because lignin, hemicellulose, pectin, and wax are eliminated. NaOH is frequently used as an alkaline treatment for fibers because it is simple, inexpensive, and efficient in enhancing filler–matrix adhesion [35].

Ahmad et al. [36] developed 0, 3, 5, and 7 wt% oil palm fiber composite filaments for FDM in 2021. The oily surfaces were cleaned from the oil palm fibers by soaking them in water for two days. The dried fiber was then treated with a sodium hydroxide (NaOH) solution for two hours to remove the hemicellulose, cellulose, pectin, and lignin. Meanwhile, Nasir et al. [32] treated a sugar palm fiber composite with alkaline and silane. The solution was exposed to the sugar palm fiber particles for three hours at room temperature. The fiber was then washed with distilled water until a pH of neutral was obtained. The fibers were heated in an oven set at 60 °C for 48 h. Natural fiber-reinforced thermoplastic that will be utilized in FDM often comes in wire form. Figure 3 shows the basic steps involved in producing a new composite material for FDM. To prepare the material in granule form for extrusion, several steps are required to create the wire filament, such as screening, crushing, mixing, compounding, grinding, and extrusion. 

Meanwhile, Xie et al. [38] created a filament using PLA and wood flour. Then, poplar wood powder was sieved to produce particles that were 140 to 160 mesh in size. In the research, they used tributyl citrate, glycerol, and distilled water. A combined amount of 630 g of PLA and 270 g of wood flour were used to make the filaments. The PLA and wood flour filaments were divided into three equal groups: 4 wt% glycerol, 2 wt% glycerol + 2 wt% tributyl citrate, and 4 wt% tributyl citrate. In a previous study, Aumnate et al. [39] developed a composite filament made of kenaf fiber for FDM. At 80.2 °C for three hours, kenaf fibers received treatment with 12% *w*/*v* NaOH in a 1:20 fiber to liquid ratio. The kenaf fibers were then dried in an oven after being treated. The dried fibers were combined with PEG to enhance the flowability of the fibers in the PLA matrix before being used to create the PLA/kenaf composite filaments. The mixture was subsequently melt compounded with virgin PLA to achieve the required final composition of 10 wt% kenaf fiber. The melt compounding method was carried out using a compounder with a screw speed of 40 rpm and a temperature of 180 °C.

Recently, researchers have been increasingly interested in creating fiber composite filaments for 3D printers. For example, Diugou et al. [40] have fabricated flax yarn-reinforced PLA filament as a material for FDM and the flax yarns were provided by SAFILIN. Meanwhile, Tao et al. [21] investigated a PLA composite made from wood sawdust. The composite filaments of 0, 2.5, 5, 7.5, and 10 wt% of kenaf/ABS have been successfully developed by Han et al. [41]. Kenaf fibers were provided by a local supplier in the form of powder. The research applied the use of kenaf fiber that was, on average, 120 µm in length. A twin-screw extruder was used to compound and extrude the mixture of kenaf fiber and ABS pellets.

Furthermore, using rice husk as filler, Pereira et al. [42] produced a natural fiber filament for FDM. The rice husks were obtained from the de-husking stage of the rice milling process. The husks were separated from any undesired fibers, grains, and contaminants as the initial step in the preparation of the fiber. The rice husks were then taken off and dried for 12 h in an oven. After drying, they were sieved through a 2 mm mesh in order to remove any impurities that remained. Later, the rice husks were pulverized into a powder with a consistency of less than 0.5 mm. To avoid absorbing moisture before alkali treatment, all the fibers were preserved in the oven at 50 °C. The rice husks were treated to an alkali treatment by being submerged in a NaOH solution. The fibers were then washed in distilled water until the pH was below 7. The fibers were then filtered and dried for 12 h at 60 °C in the oven.

## 3. Extrusion Process

### 3.1. Single-Screw Extruder

Typically, the extrusion process is used to fabricate wire filaments for FDM. Figure 4 depicts an illustration of a single-screw extruder that is currently on the market. The single-screw extruder was invented in the 1870s. Due to its simplicity of use in the manufacturing of polymers and rubber, it is the most widely used extruder [43]. Processes such as sieving the fibres, drying, and mixing techniques have a direct impact on the properties and acceptability of the manufactured filaments. To ensure that the fibre and polymer matrices are well bonded, it involves drying and surface preparation. The single-screw extruder design consists of one rotating screw that rotates inside a static tubular barrel that is divided into three zones: compression, feed, and metering [44]. Because of the increasing resistance and thermal energy as the screw speed rises, single-screw extruders are inappropriate for heat-sensitive materials. Moreover, they lack the ability to combine enough materials to create a polymer composite employing numerous compounded components. They basically have one operating screw system and are employed to produce homogenized polymers with continuous shapes [45].

The barrel pressure increases gradually along the length of the compression zone when the screw flight and pitch depth are reduced [47]. By adjusting the depth and pitch of the screw flight within each zone, different pressures can be produced along the length of the screw. To ensure low pressure in the feed zone, which effectively feeds material from the hopper into the extruder barrel, the screw flight pitch and depth are often selected at larger scales than from other zones [48]. A single-screw extruder is especially suitable for producing extrusion-based parts because it can operate at high pressure, which is ideal for high viscosity polymeric materials [49]. Furthermore, during the extrusion process, a substantial amount of pressure is applied, compressing the materials to create filaments. Due to a lack of shear deformation, it can potentially result in agglomeration and insufficient mixing [50].

Table 2 introduces the FDM natural fiber-based filaments obtained through the extrusion process. The following are the researchers who developed the natural fiber-based filament for FDM: Ahmad et al. [51], Singh et al. [52], Han et al. [41], Girdis et al. [53], Osman et al. [23], Nafis et al. [54], Milosevic et al. [24], Scaffaro et al. [55], Figueroa et al. [56], Shahar et al. [57], Jamadi et al. [58], Liu et al. [39], Yu et al. [59], Fekete et al. [60], Depuydt et al. [61], Tao et al. [21], and Dey et al. [62]. The common natural fibers used in fabricating the filament are oil palm, banana, kenaf, nutshell, rice straw, wood, hedysarum, agave, sugarcane, astragalus, and bamboo. Table 2 contains the data of the extrusion process, the 3D printer model, and the status of printability. The most popular brand of extruders used in the current research are LabTech, Haake Technik, Wellzoom, and Leistritz. In a recent study, Tao et al. [41] developed the wood flour/PLA filament for 3D printing. To produce wood flour/PLA composite filaments, they employed a single-screw extruder (Wellzoom) with parameter settings of 2 m/min^−1^ speed and a 175 °C barrel temperature. To remove moisture, the composite granules were first warmed for four hours at 103 °C. Subsequently, Pareira et al. [42] produced the rice husk composite filaments using a single-screw extruder (3Devo) with a screw speed of 4 rpm and barrel temperatures of 175, 180, 190, and 180 °C for four zones. To produce compositions of 5, 10, 15, and 20 wt%, the PLA granules were mixed with rice husk after being dried at 80 °C for 4 h in an oven. 

### 3.2. Twin-Screw Extruder

To manufacture similar mixtures of two or more distinct materials, a twin-screw extruder with two screws positioned next to each other at a modular barrel was created in 1930 [63]. A twin-screw extruder was developed to manufacture a thermoplastic filament made of natural fibers that enables for light scattering and distributive mixing. The polymer matrix and natural filler should be combined, which can be accomplished by compounding the fibers and polymer using a twin-screw co-rotator. The materials are properly mixed as a result of the increased shear stress between the rotating screws and barrel provided by this device [64]. In addition to preventing excessive overheating of raw materials, the intermeshing twin-screw extruder can also eliminate non-motion during extrusion. Even after operation, the screws are rotated to clear the inside of the barrel and eliminate any leftover debris from the screw roots [65]. Examples of natural fiber extrudate filament (wood, kenaf, and astragalus) for FDM are shown in Figure 5.

Xie et al. [38] used a twin-screw extruder (SHJ-20) to manufacture the wood/PLA filament. Every section had its extrusion temperature set to 135 °C, 150 °C, 170 °C, and 135 °C, respectively. Moreover, Liu et al. [22] used a twin-screw extruder (Haake) to create a composite filament made of sugarcane and PLA for FDM. Fiber and PLA granules were dried at 60 °C for 10 h before extrusion. The temperatures for extrusion were set at 170 °C to 195 °C from the hopper to the die. Extrudate filament had a diameter of 1.75 ± 0.05 mm, and the screw speed was set to 50 r/min. Meanwhile, Han et al. [41] used a twin-screw extruder (HTGD 20) with a die diameter of 1.75 mm to develop a kenaf/ABS composite filament. Nafis et al. [54] also used a twin-screw extruder to produce a recycle PP/wood composite filament. The diameter of the extruded filament was between 1.5 and 1.79 mm. The screw speed for the extrusion process was set at 42 rpm, and the barrel temperature was set to 160–200 °C. The barrel temperature is crucial for producing composite filament of high quality. The physical properties of the extruded filament would be affected by entering the incorrect barrel temperature. The barrel temperatures of natural fiber-based filaments in previous studies are shown in Figure 6. Based on these results, the temperature range for thermoplastic reinforced with natural fibers was 160 to 250 °C. The ABS-mixed fiber mixture was at a higher temperature setting because ABS has a higher melting point. The feed zone, transition zone, and melting zone are the three heating zones in the barrel [66]. Using a hopper, the mixture of composite materials is injected into the extruder’s barrel. The rotating screws are enclosed in the barrel. In the barrel, the raw materials are also heated. The feed zone softens the composite mixture, the transition zone plasticizes it, and the melting zone totally dissolves it. Granulated composite materials will pass through the revolving screws’ surface from the feed zone to the transition zone, then to the melting zone. Several zones have varying temperatures, which are chosen based on the composite materials. In a recent study by Dey et al. [62], they manufactured a PLA/soybean filament using a twin-screw extruder. In order to set the barrel temperature, the twin-screw extruder barrel’s multiple zones were heated to a temperature ranging from 138 °C to 171 °C. The molten composite was forced from the melting zone through a strand die once it has melted entirely. The continuous composite string was then fed through a 90 °F water bath.

## 4. Dimensional Stability and Printability

The accuracy of the extrudate filament’s diameter is crucial to the FDM printing process. A number of factors, such as material processing conditions, the uniform particle distribution of the filler and adhesive materials, and rheological behavior, determine whether a composite material feedstock is acceptable for the FDM process [51]. The filaments used in FDM methods typically have a 1.75 mm diameter. The diameter ranges of natural fiber-based FDM filaments are shown in Figure 7. The diameter was found to be between 1.54 and 1.9 mm. The majority of the composite filaments’ diameters were only slightly larger than the nominal size (1.75 mm), except the maximum values for the composite filaments made of PP/wood [54] and PP/hemp [24] were outside of tolerance.

Since FDM 3D printers rely entirely on software that assumes a constant diameter of the filament, the filament’s diameter must remain constant or without significant variations. The quality of the printed specimen is affected by under- or over-extrusion caused by filaments that are too small or large, respectively [67]. The non-uniform fiber distribution, coil speed, non-uniform composite particle size, and rheological properties of the material could all have an effect on the extruded filaments’ dimension stability. Gkartzou et al. [68] developed a composite filament made of wood and PLA. It was discovered that the measured filament diameters have a normal distribution with a standard deviation of 0.02 mm, which indicates that 95% of the filament utilized has a tolerance of 0.04 mm. During the melting and deposition of the materials, a close diameter tolerance is crucial.

Despite this, there was not much information in the literature regarding the way researchers ensure the filament is extruded with uniform dimensions. However, there is still minimal research regarding the topic of using NFRC for AM technologies, especially FDM 3D printing technology. This demonstrates that there remains a lot of potential for investigation in this field of study.

## 5. Characterization of Wire Filament

### 5.1. Mechanical and Physical Properties

To evaluate the strength of extrudate filaments, it is recommended to perform a wire pull test in order to analyze mechanical performance, such as tensile strength and tensile modulus. A material’s tensile strength can be used to measure how much strain it can sustain or how much it can stretch before failing. A strain gauge was attached to the sample at the focus point in order to perform the tensile test. The results of previous studies on the density and porosity of natural fiber-based filaments for FDM, as well as the results of the wire pull test, are summarised in Table 3. According to Antony et al. [69], a paper frame method was used to test the tensile properties of hemp/PLA filaments. The filament was found to have a tensile modulus of around 1530 MPa and a tensile strength of approximately 28 MPa. Meanwhile, Shahar et al. [57] revealed that as fiber loading increases, the tensile strength starts to decrease. This is because of the poor bonding between the filler and the matrix. Furthermore, Nafis et al. [54] reported that wood dust filament treated with silane had the highest wire tensile strength compared with all the other filaments. Filaments made from treated wood (silane) had a strength that was 35.4% more than that of untreated wood. The lowest void and gap, which are seen in the morphology of the threads’ fracture surface, are accountable for the material’s maximum strength. The filament with the fewest flaws will be able to carry a heavier load and will thus be stronger.

To test the porosity of the ABS/nutshell composite filament, Girdis et al. [53] applied an Archimedes principal in their study. The densities of the nutshell composite samples were significantly lower than the commercial wood and neat ABS filaments. On the other hand, Depuydt et al. [61] used computed tomography to measure the density and porosity of the extrudate filament (PLA/bamboo). All the extrudate filaments were less porous than the commercially available filament. Furthermore, Shahar et al. [57] used a densimeter to measure the density of a PLA/kenaf composite filament. The density of the specimen decreases as the amount of filler increases. In conclusion, the density of the natural fiber-reinforced thermoplastic filaments were lower than the neat thermoplastic filaments. This is because the presence of fibers in the matrix composite makes them porous. 

### 5.2. Morphology and Surface Quality

A morphological study of the extrudate composite filament is critical, as it is used to examine the dispersion of fibers in the matrix and observe the surface quality of the filaments. Ahmad et al. [51] observed the extrudate filaments that were fabricated from an oil palm fiber composite and found that the dispersion of fiber particles is heterogeneous, influencing the specimens’ structural strength and mechanical performance. The fiber had not completely dissolved into the matrix, which later caused problems with 3D printing. Furthermore, it was causing pores that weaken the interfacial relationship between the fiber and the matrix composite. These pores are a consequence of the intense spinning, which is brought on by the fountain flow phenomenon [70]. Microscopic images of extrudate filaments (5 wt% pine/PLA and neat PLA) are shown in Figure 8. It has been demonstrated that fiber, when compared with neat PLA, significantly increases the surface roughness of fibers. In the extrusion process, complex heat and mass transfer mechanisms, along with the buildup of mechanical and thermal stress and phase shifts, result in the development of bonds between individual fibers. The expansion of the neck created between nearby fibers, as well as molecular diffusion and randomness at the interface, all affect the strength of these interactions [71].

In a recent study, Muck et al. [72] developed a wood/PLA composite filament and found that the outer surface of the filament has a rough surface and a highly porous structure, as in Figure 9a,b. The filament has fiber pullouts, gaps between the fiber and the matrix, an extremely uneven transverse surface and structure, and a non-eccentric diameter. This is because there is insufficient interfacial adhesion between the PLA matrix and the wood fibers. As reported by Duigou et al. [73], the wood-based filament with a porous structure will contribute to water transport and react as a hygroscopic characteristic. 

## 6. Challenges in Developing a Fiber Composite Filament

There were some challenges during the development of a fiber-reinforced thermoplastic composite, especially for FDM 3D printing. Natural fiber-based composite filaments for FDM are prepared by the single-screw extruder or twin-screw extruder. Nevertheless, these have produced a number of difficulties, including an uneven mixture of the fillers and the polymeric matrices, temperature regulation, and the development of holes throughout production [3,61,74]. In the meantime, they lead to issues such as changing mechanical properties and nozzle clogging in 3D printers [74]. Chemical treatments such as crosslinking agents [3], bonding agents [75], and chemical techniques [22] are added to the solution containing the polymeric matrix and fillers in order to increase the interface adhesion between the matrix and fillers. According to Petchwattana et al. [76], only composite materials produced with a 70 µm particle diameter can be printed successfully. The polymer matrices and fibers are combined once the polymer and fibers have undergone complete processing. For natural fiber-reinforced polymer composites to be used in 3D printing, particularly in the FDM process, the size of the fiber particles must be as small as possible in order to prevent contamination at the printing nozzles. The powders may occasionally be screened based on the source of the fiber in order to ensure that no bigger particles are mixed with the composites. This stage is important because what happens next will have an immediate effect on the filament quality. Moreover, Guen et al. [77] claimed that these findings were obtained using a variety of combination approaches, including first blending the dryness, then mixing the melt, and lastly vigorously extruding the filaments.

According to Ahmed et al. [37], when using natural fibers, the processing needs to be undertaken diligently, or else it may lead to low quality filament or poor outcomes. These fibers must be dried carefully in the initial phase even before compounding as it is very important to reduce the water content in them, which, if not performed correctly, could lead to hydrolytic degradation. The temperature used should be monitored carefully to avoid thermal degradation. Meanwhile, Shahar et al. [57], while fabricating a PLA/kenaf composite filament using a single-screw extruder, found that the extruding pellets would begin to gather at the extrusion nozzle if the temperature were too low, resulting in blockage and a forced halt to the extrusion operation. Moreover, the buildup of fibers close to the die head may result in irregular extrusion flow, which would produce filament with a non-stable diameter. Hwang et al. [78] discovered that polymer composite filaments tend to increase as the temperature decreases. The PLA had portrayed this state in a straightforward manner. Since the filament was manufactured entirely of thermoplastic, it contained no additional materials that might obstruct the extrusion process. The texture of the filaments continued fine and devoid of bumps regardless of the level of temperature utilized during extrusion, demonstrating that the filaments continued to flow freely despite the slowing extrusion flow. Additionally, a lack of adhesion between the fibers and the polymer matrix causes problems when making a homogeneous blend. George et al. [79] found that the fibers could frequently aggregate and form bulges in the polymer matrix. Agglomeration is an issue when creating feedstock for AM due to frequent blockages [80], regardless of the claimed major improvements in the characteristics of nanocellulose materials. However, the main challenges with nanocellulose materials continue to be the removal of various components to enhance their performance and recycling issues [81]. To solve the current challenges, future research on nanocellulose materials must take a variety of factors into account, such as machine learning, artificial intelligence, and numerical models.

## 7. Conclusions and Future Perspectives

In conclusion, rigorous procedures and a thorough understanding of the characteristics of natural fiber and its matrix (thermoplastic) are required for the development of natural fiber composite filaments for FDM. A rheological study is also important before starting the extrusion process for extruding new filament wires made from natural fiber. It helps in identifying the appropriate melting temperature and melt velocity. The density of the natural fiber-reinforced thermoplastic filaments were lower than the neat thermoplastic filaments. This is because the presence of fibers in the matrix composite makes them porous. It was also discovered that increasing the fiber content caused the density to decrease but the strength to significantly increase.

For future perspectives, the mechanical properties of extrudate composite filaments have been improved compared with pure thermoplastics, as natural fiber composite materials are sustainable, compactable, and biodegradable, in the field of FDM applications. Furthermore, natural fiber-reinforced thermoplastics are less harmful to the environment when compared with thermoplastic materials that release hazardous gases. Before the newly developed natural fiber-containing 3D-printed composites can be used in industry, advancements in the procedure, materials, characteristics, and structures are required. The selection of materials for natural fiber composites required a focus on functional qualities as well as mechanical behaviors, such as the degradation rate for environmental friendliness, porosity for cushioning, conductivity for self-detecting, and hydrophobicity for flammable cladding. Prior to printing, it is crucial to do a careful study of the filament’s composition and quality by selecting the proper extrusion process parameters, such as the correct screw speed, spooling speed, barrel temperature, pressure, etc. Further research on natural fiber-filled polymers is still necessary. To further minimize the environmental impact, emphasis should be placed on using industrial waste to produce bio-based thermoplastic composites.

## Figures and Tables

**Figure 1 materials-16-04052-f001:**
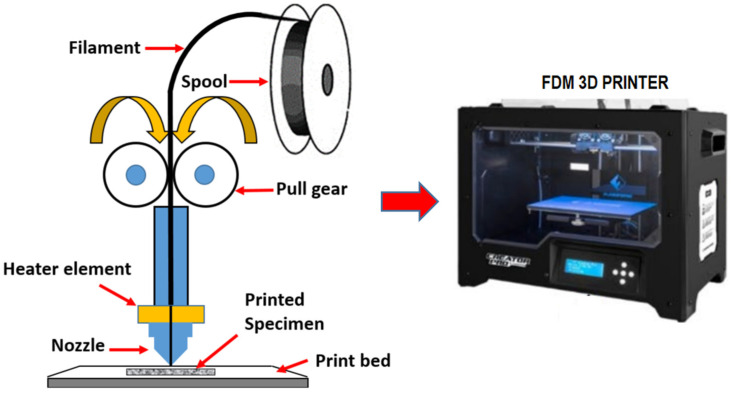
Schematic illustration of a FDM 3D printing [19]. Licensed under creative commons attribution 4.0 international license.

**Figure 2 materials-16-04052-f002:**
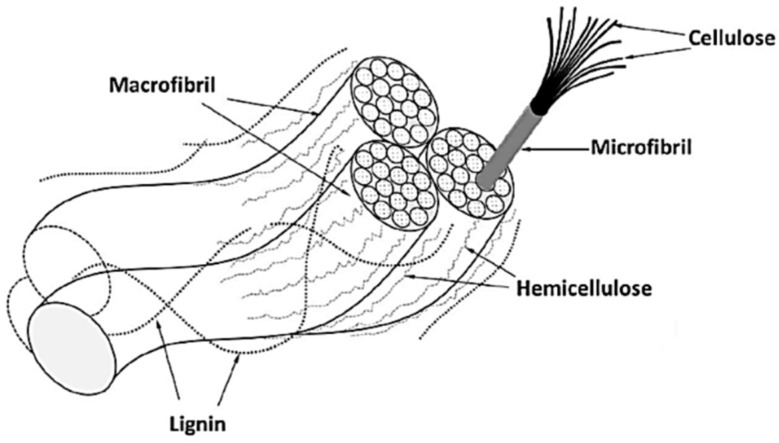
Graphical image of natural fiber composition [32]. Licensed under creative commons attribution 4.0 international license.

**Figure 3 materials-16-04052-f003:**
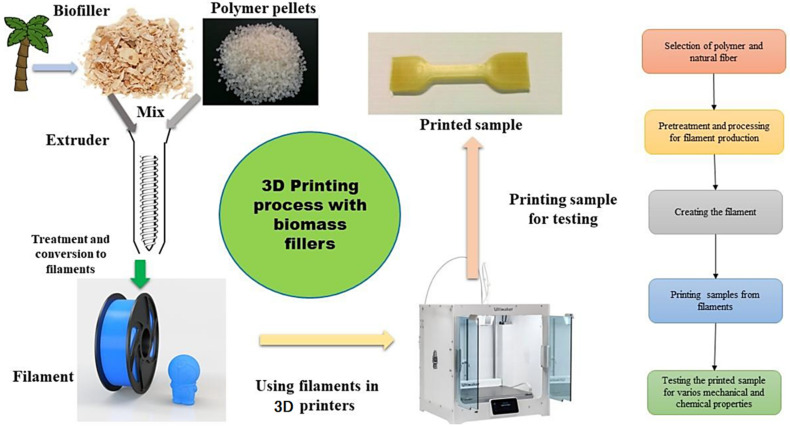
Process of making the natural fiber filament for FDM and printed sample [37]. Licensed under creative commons attribution 4.0 international license.

**Figure 4 materials-16-04052-f004:**
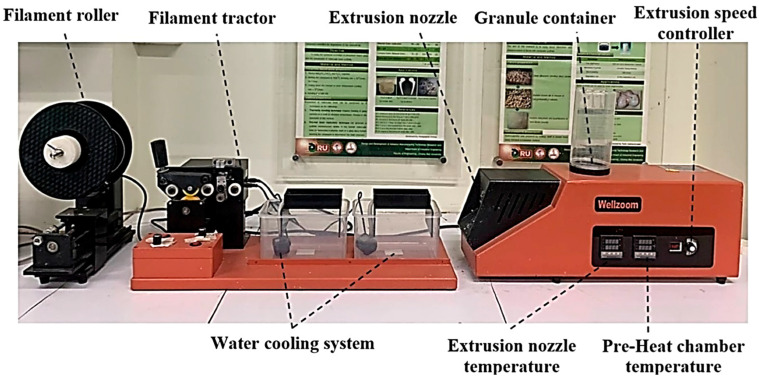
Single-screw extruder for making the fiber composite filament [46]. Licensed under creative commons attribution 4.0 international license.

**Figure 5 materials-16-04052-f005:**
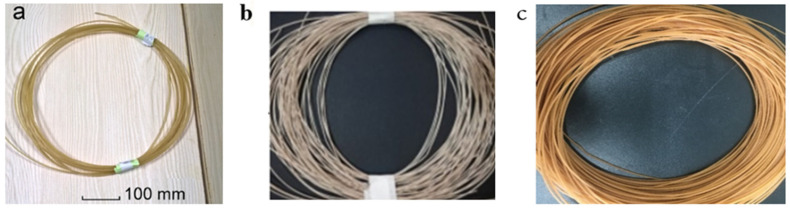
Extruded natural fiber composite filament (**a**) wood/PLA [21], (**b**) kenaf/ABS [41], and (**c**) astragalus/PLA [59] composite filament. Licensed under creative commons attribution 4.0 international license.

**Figure 6 materials-16-04052-f006:**
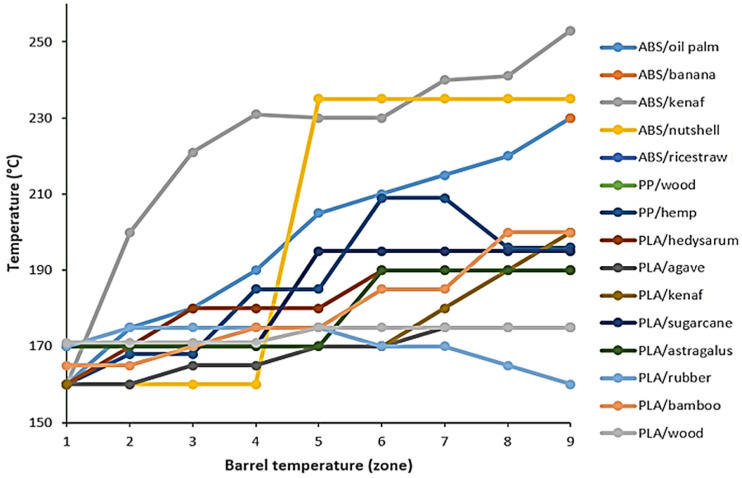
Barrel temperatures of natural fiber-based filaments include PLA/wood [21], ABS/rice straw [23], PP/hemp [24], PLA/sugarcane [39], ABS/kenaf [41], ABS/oil palm [51], ABS/banana [52], ABS/nutshell [53], PP/wood [54], PLA/hedysarum [55], PLA/agave [56], PLA/kenaf [57], PLA/astragalus [59], PLA/rubber [60] and PLA/bamboo [61].

**Figure 7 materials-16-04052-f007:**
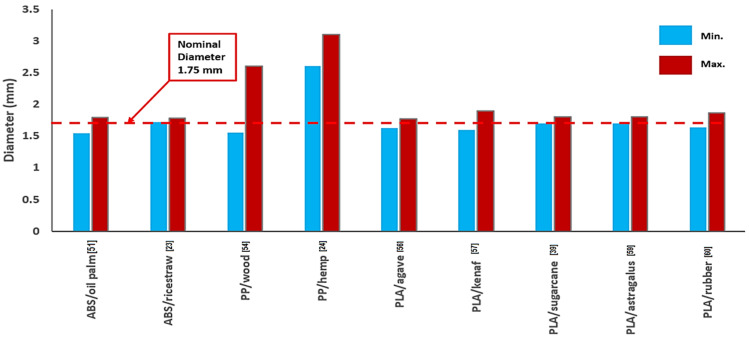
Diameter ranges of natural fiber-based filaments for FDM.

**Figure 8 materials-16-04052-f008:**
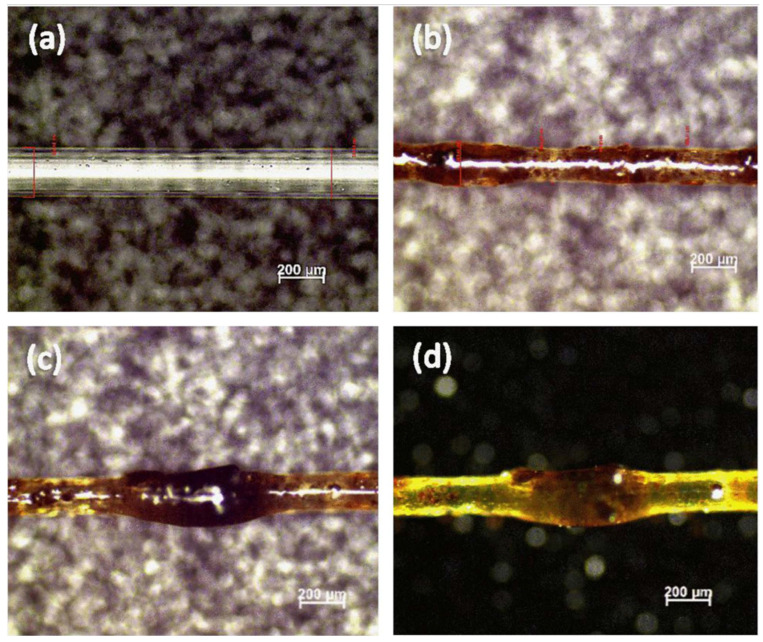
The microscopic images of extrudate filaments (**a**) neat PLA and (**b**–**d**) 5 wt% of pine lignin/PLA with different speed of extrusion [68]. Licensed under creative commons attribution 4.0 international license.

**Figure 9 materials-16-04052-f009:**
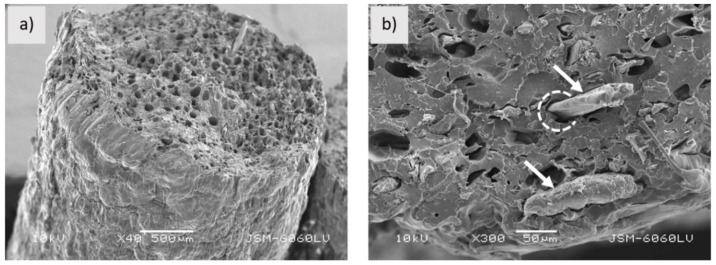
SEM images (cross sectional view) of wood/PLA filament (**a**) magnification 500 µm and (**b**) magnification 50 µm [72]. Licensed under creative commons attribution 4.0 international license.

**Table 1 materials-16-04052-t001:** The mechanical properties of natural fibers [31,32].

Fiber	Density (g/cm^3^)	Tensile Strength (MPa)	Tensile Modulus (MPa)	Elongation at Break (%)
Sugar palm	1.30	15.5–290	0.5–3.4	5.7–28
Sisal	1.50	400–700	9.0–38.0	2.0–14.0
Oil palm	1.55	400	9.0	18.0
Jute	1.60	393–800	10.0–30.0	1.2–1.8
Kenaf	1.45	930	53.0	1.6
Hemp	1.48	550–900	70.0	1.6–4.0
Cotton	1.60	287–800	5.5–12.6	2.0–10.0
Bamboo	1.25	290	17.0	-
Flax	1.50	345–1500	27.6	1.2–3.2
Pineapple	1.44	413–1627	60.0–82.0	14.5
Banana	1.35	529–914	27.0–32.0	5.9

**Table 2 materials-16-04052-t002:** Extrusion process and printability of natural fiber-based filament for FDM.

Matrix	Filler/Reinforcement	Wt.%	Extruder (Brand/Type)	Die Diameter (mm)	Filament Diameter Range (mm)	Printability (Yes/No)	3D Printer	Ref.
ABS	Oil palm fiber	0–7	E23 Siemens/Twin screw	1.75	1.54–1.79	Yes	Flash Forge Creator Pro	[51]
ABS	Banana fiber	0–5	Customized/Twin screw	1.50	-	Yes	-	[52]
ABS	Kenaf fiber	0–10	HTGD-20/Twin screw	1.75	-	Yes	FlashForge	[41]
ABS	Macadamia nutshell	0–29	Customized/Single screw	-	-	Yes	Leapfrog Creatr	[53]
ABS	Rice straw	0–15	Customized/Single screw	1.75	1.72–1.78	Yes	Printrbot Simple Metal	[23]
PP	Wood Dust	0–3	Lab Tech/Twin screw	1.75	1.55–2.60	-	-	[54]
PP	Hemp	0–30	Lab Tech Scientific LTE20-44/Twin screw	3.00	2.60–3.10	Yes	Diamond Age	[24]
PLA	Hedysarum coronarium	0–20	Haake Technik/Single screw	1.75	-	Yes	Sharebot, Next Generation	[55]
PLA	Agave fiber	0–10	Leistritz Micro/Twin screw	1.75	1.63–1.77	Yes	Wanhao Duplicator	[56]
PLA	Kenaf fiber	0–7	Well zoom/Single screw	1.75	1.60–1.90	Yes	FlashForge Creator Pro	[57]
PLA	Kenaf fiber	0–2.5	Customized/Twin screw	1.75	-	Yes	FlashForge Creator Pro	[58]
PLA	Sugarcane fiber	0–15	Polylab OS/Twin screw	1.75	1.70–1.80	Yes	Architect 3D	[39]
PLA	Astragalus	0–15	SHJ-20, Nanjing Giant Machinery/Twin screw	1.75	1.70–1.80	Yes	Moshu S108	[59]
PLA	Rubber	0–20	Labtech LTE 20–44/Twin screw	1.75	1.64–1.86	Yes	Craftbot Plus	[60]
PLA	Bamboo	0–15	Leistritz ZSE/Twin screw	-	-	Yes	Low cost 3D printer	[61]
PLA	Wood	0–5	Wellzoom LLC/Single screw	1.75	-	Yes	Shenzhen 603S	[21]
PLA	Soybean hulls	0–10	Leistritz/Twin crew	1.75	1.74–1.76	Yes	MakerBot Replicator Z18	[62]

**Table 3 materials-16-04052-t003:** Result of wire pull test, density, and porosity of natural fiber-based filament for FDM.

Matrix	Filler	Wire Pull Test/UTM	Density/Porosity Test	Results	Ref.
ABS	Oil palm fiber	Shimadzu Autograph (AGSX), ASTM638	Archimedes principle (ASTM D3800)	Tensile strength was increased by 60% by going from 0.15 to 0.4 MPa of fibre loading. After that, the Young’s modulus increased by 22.8%, from 16.1 to 18.3 MPa. As the fiber loading was increased from 3 to 7 wt%, the density of extruded filament decreased and the percentage of porosity rose.	[51]
ABS	Nutshell	-	Archimedes principle	The two nutshell composite samples’ densities are much lower than those of pure polymer filaments and commercial woodfill. It is clear that when 29% nutshell is added, the density is reduced by more than 27.4% compared with pure ABS.	[53]
PP	Wood dust	AI-7000-LAU Go-Tech, ISO 3341	ASTM D792-91	According to the results, treated silane has a filament strength that is higher than r- PP’s. The findings show that the silane pretreatment of the wood fiber improves the interaction between it and the recycled PP.	[54]
PLA	Kenaf	Instron 3382	Densimeter (Mettler Toledo)	The sample becomes less dense when the filler content rises from 3 to 7 wt%. Tensile strength starts to decline as fiber loading rises. That is as a result of the filler and matrix’s insufficient bonding.	[57]
PLA	Bamboo	Instron 5567	Computed tomography (Phoenix Nanotom 180)	The compounded fibers’ length to diameter ratio had an effect on the filament’s modulus. Long bamboo fibers added to the PLA filament enhanced its stiffness by 215%, compared with a 39% rise for the dust-like fractions. Less porosities are present in every filament developed for this study compared with commercial filament. The porosities can be decreased to levels between 0 and 4% by anticipating appropriate drying.	[61]
PLA	Hemp	Instron 4484	-	The filaments initially act elastically up to 18–20 MPa, but as the test goes on, the filaments behave viscoelastically. Tensile modulus is between 1500 and 1575 MPa, and it breaks between 25 and 30 MPa.	[69]

## Data Availability

Data will be made available on request.

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
