# Peer review of "A Review of Natural Fiber-Based Filaments for 3D Printing: Filament Fabrication and Characterization"

_materials, 2023, doi:10.3390/ma16114052_

Round 1

Reviewer 1 Report

Additive manufacturing has been a hot topic in recent decades, and lots of works has been done. The authors reviewed the filaments for the fused deposition modeling technology. The natural fibers have been used as bio-filters with thermoplastics as the filament. This work presents a comprehensive review of the natural fiber-based filaments, including the fabrication process, characterization method, and physical properties. The manuscript is well-structured and has enough content. It also fits for the materials journal. There are just a few issues that should be addressed before the acceptance. Please see comments below.

1.      Introduction section, the authors should briefly introduce the differences between various AM technology, instead of just saying their names. The advantages and disadvantages should also be added.

2.      Line 81 and 82. ‘composite composites’.

3.      Line 141, ‘80 2 °C’?

4.      Line 213, ‘175, 180, 190, and 180°C’. What does this mean?

5.      Figure 5b has a poor quality.

There are a few typos in the text, which should be paid attention to. 

Author Response

Dear reviewer,

The authors would like to thank the reviewer for his/her constructive comments and suggestions that would improve the quality of the paper. The responses to the comments are in the attachment. 

Thank you.

Reviewer 2 Report

The authors reviewed recent studies on the 3D printable filament using natural resources. The review has its value since the sustainability and 3D printing is important. There are several minor issues that should be addressed.

1. The authors discussed several resources of the fibers. However, direct use of cellulose fibers and characterization of cellulose fibers are not well discussed. Several publications are recommended below:

A. Islam, M. N. & Jiang, Y. 3D Printable Sustainable Composites with Thermally Tunable Properties Entirely from Corn-Based Products. ACS Sustain. Chem. Eng. 10, 7818–7824 (2022).

B. Li, T. et al. Developing fibrillated cellulose as a sustainable technological material. Nature 590, 47–56 (2021).

C. Siqueira, G. et al. Cellulose Nanocrystal Inks for 3D Printing of Textured Cellular Architectures. Adv. Funct. Mater. 27, 1604619 (2017).

2. Several minor typos should be corrected, including Page 2 line 81 ”… found that composite composites that used natural…”, Page 4 line 120 “…The undesirable soluble substances such as hemicellulose, cellulose, pectin, and lignin were then removed…”.

3. In Figure 6, the 9 zones are not clearly defined to the reviewer. Also, how the data was collected from the literatures in this figure is unclear. 

Several typos are found as listed above. Please go through the manuscript carefully to avoid them.

Author Response

(The authors gave the same response as above.)

Reviewer 3 Report

In this Manuscript entitled “A review of natural fiber-based filaments for 3D printing: filament fabrication and characterization”, the authors presented the fabrications methods and characterization of thermoplastic material blended with natural fiber produced wire filament. Moreover, the characterization of 3D printing feedstock incorporated mechanical properties, dimension stability, morphological study, and surface quality.

Overall, some major issues are associated with this review article, which needs to be addressed before possible publication in Materials.

Please find the attached annotated file to see my comments.

Lastly, I would like to say “Materials” journal publishes high-quality research articles related to biocomposites. Based on my comments mentioned in annotated file, the recommendation is Major Revision.

The manuscript requires a thorough language editing

Author Response

Dear reviewer,

The authors would like to thank the reviewer for his/her constructive comments and suggestions that would improve the quality of the paper.

Please refer to the revised paper (in red color), where all errors have been fixed.

Thank you.

Reviewer 4 Report

1. FFF (fused filament fabrication) can be mentioned, which is used on a par with FDM.

2. Since the authors motivate the use of natural fibers by the impact on the environment, it would be interesting if they considered or mentioned technologies for the disposal / recycling of composites containing natural fibers.

Author Response

(The authors gave the same response as above.)

Reviewer 5 Report

The manuscript "A review of natural fiber-based filaments for 3D printing: filament fabrication and characterization” presents an overview of natural fiber-based filaments for 3D printing. Due to the significant advances of 3-D printing techniques, the topic is of real interest for readers. The authors structured their review starting from the material types used and their preparation, continuing with the extrusion type processes and characteristics. 

In the following are summarized some observations.

1. The keywords list should be modified to reflect properly the contents. For example, is not suitable to put "characterization". 

2. The Introduction Section should be completed.

3.Line 50 - Reformulate including an appropriate reference. It would sound better as: "The FDM technology was invented in 1989 [Reference]."

4. Line 98 - delete “. in the paper."

5.  In the Section Material Preparation - Line 138 - What relevance has the fact that the chemicals were imported from China? Please reformulate.

6. Line 139 - The info that "three equal groups." is just left unfinished - what represents these three groups, how were used, and so on. Please complete the necessary info.

7. Line 140 - "Additionally" is not adequately used here. Revise.

8. Line 149 -."trend of researchers." What does it mean? Revise and reformulate.

9. Section 2 - Material Preparation - the authors should reconsider the text and put their info into a more comprehensive and systematic presentation. For example, to consider each type of natural material used for filaments and to present them separately into a subsection. Then, it could be introduced details regarding the preparation procedures and consequently many other references. 

At this point, the review lacks rational and systematic approach of presentation.

10.  Line 171 - "The main method for producing." - as it is formulated the phrase, it signifies that there are some other methods as well, but the authors made no reference to them. Please complete.

11. Line 199 to Line 203 - Delete the phrase and reformulate it, as all references are also included in Table 2.  For example, as: "Table 2 introduces the FDM natural fiber-based filaments obtained through the extrusion process".

12. Line 206 - 207 - The info was already included in the table. Delete it. 

13. Line 248-249 - Figure 6 - is it an original graph? It is not clear. If it is reproduced from another paper that it should be completed with the reference and the publishing agreement.

14. Line 275-276 - Figure 7 - same comments as Figure 6. If the graph is original, then the references from where the data were taken should be mentioned.

15. Line 371 - There is mentioned that the fiber composite filaments are "mainly" prepared through extrusion process. That "mainly" indicates that there are some other preparation alternatives. If so, and this is the case, the authors should present these alternatives as well. 

16. Line 428-429 - Do the authors refer to only one parameter to be adjusted for a preparation extrusion process??? Please complete adequately the phrase.

17. The References Section should be completed.

There are some valuable reviews on the topic that the authors did not mention, but such previous works should be considered.  For example:

- Xiaoyu Bi, Runzhou Huang, 3D printing of natural fiber and composites: A state-of-the-art review, Materials & Design, Volume 222, 2022, 111065, ISSN 0264-1275, doi: 10.1016/j.matdes.2022.111065.

or 

- Rajendran Royan, N.R.; Leong, J.S.; Chan, W.N.; Tan, J.R.; Shamsuddin, Z.S.B., Current State and Challenges of Natural Fibre-Reinforced Polymer Composites as Feeder in FDM-Based 3D Printing, Polymers 2021, 13, 2289, doi: 10.3390/ polym13142289

The English Language should be carefully revised.

Some suggestions:

- Line 94 - reformulate the phrase

- Line 95 - ... that was made ..."

- Line 100 - better use: ".. are presented..." instead of .."were also provided.."

- Line 110-111 - necessary to reformulate the phrase

- Line 127 - reformulate the phrase

- Line 172-173 - reformulate the phrase

- Line 409 - reformulate the phrase

- Line 421-422 - reformulate the phrase for a correct understanding

Author Response

(The authors gave the same response as above.)

Reviewer 6 Report

The manuscript summarised the information of the 3D printing filaments with respect to the most commonly used materials such as ABS, PP, and PLA. In the most general sense, it describes the basics and essence of 3D printing from historical aspects to the principal arrangement of the technology.

The majority of the manuscript focusses on natural additives/natural fibres (as are sugar palm, sisal, jute, cotton, bamboo, or banana), which are used as modifiers of polymer matrices. The effects of natural additives on changes in mechanical properties, dimensional stability, morphological properties, and surface quality are monitored.

The manuscript is prepared in a very readable and clear form. In the form of a review, it summarises findings that have significant potential in the future application of ecological materials to reduce the costs of producing polymer materials for 3D printing filaments and their recycling. The authors draw, among other things, from their own scientific knowledge and experience, where they cite 8 scientific publications in the case of author Mastur Mohammad Taha and 5 scientific publications in the case of Zulkiflle Leman.

I have no comments on the manuscript and I recommend publishing it in the present form. Please only correct the typo in picture No. 3 - 3d printers on 3D printers.

Author Response

The authors would like to thank the reviewers for their constructive comments and suggestions that would improve the quality of the paper. The responses to the comments are in the attachment. 

Round 2

Reviewer 3 Report

Accept

Reviewer 5 Report

Dear authors, thank you for the careful review of the manuscript.

The paper was significantly improved. Although, in my opinion, the Material Section should be otherwise structured and developed.

Also, I expected more references to be included.

The manuscript still needs thorough proofreading.

Few examples:

- Line 115 - "This article reviews ...." is more appropriate than "reviewed."

- Line 116 - "It discusses ..." and not "discussed."

- Line 117 - "...wire filament that fabricated from thermoplastic material..."  should be   "...wire filament made of thermoplastic material...".

- Line 120 - "... are also discussed." and not "were discussed."

- Line 121 - " ...are also presented." and not "were presented."

In the meantime, in the consecutive phrases from lines 119-122 used "also" - it is suggested to use a synonym and not to repeat the same word or expression too often.